

# Extratropical cyclones drive the spatial heterogeneity distribution of Sea Salt Aerosol (SSA) and vertical transport in the Southern Ocean

Xiaoke Zhang[1,2], Rong Tian[1,2], Jinpei Yan[1,2], Shanshan Wang[1,2], Shuhui Zhao[3,] Hanyue Xu[1,2], Qisheng Zeng[1,2], Heng Sun[1,2], Xia Sun[1,2]

1 Key Laboratory of Global Change and Marine-Atmospheric Chemistry, Xiamen 361005, China

2 Third Institute of Oceanography, Ministry of Natural Resources, Xiamen 361005, China

3 School of Tourism, Taishan University, Tai'an, China

*Correspondence to*: Jinpei Yan (jpyan@tio.org.cn)

**Abstract.** Sea salt aerosols (SSA) were emitted via bubble bursting during wave breaking, exhibiting a strong positive correlation with wind speed. However, the generation and emission of SSA driven by cyclones was still lack of knowledge. In this study, we combine cruise observations and GEOS-Chem simulations to investigate the contribution of extratropical cyclones to SSA dynamics. During the R/V Xuelong cruise, observed SSA concentrations were consistently lower in cyclonic periods compared to non-cyclonic periods, a pattern probably linked to updraft transport within cyclone systems. Model results revealed that SSA concentrates predominantly north of cyclone centers. As altitude increases, these high-concentration zones gradually shifted northwestward. Cyclone-associated high-wind regions accounted for 63 % of total SSA emissions across the Southern Ocean. The maximum upward SSA transport flux occurred at 450 m altitude within Warm Conveyor Belt regions, with stronger and longer-lasting cyclones generating greater transport intensities. Our results demonstrate that cyclones modulate SSA distribution primarily through turbulent mixing, with synergistic effects from wet deposition and advective transport. This study highlights the spatial heterogeneity of SSA distribution during cyclones and elucidates how combined physicochemical processes regulate SSA transport under cyclonic conditions.

## 1 Introduction

Sea Salt Aerosol (SSA) is an inorganic salt aerosol with sodium chloride as the main component, which is produced by the wind on the surface of the sea (Alroe et al., 2020; Quinn et al., 2015; Vignati et al., 2010). It is estimated that the global flux of SSA is around 1,000-10,000 Tg a$^{-1}$ (Gong et al., 2003). SSA is the largest aerosol in the global ocean boundary layer, with mass concentrations typically between 0.3



and 3 μg m⁻³ (Thomas et al., 2022). Although the aerosol content in the atmosphere is minimal, it plays a significant role in the global system. When cloud liquid water was constant, the increase of SSA mass concentration would reduce the effective radius of cloud droplets, thereby increasing reflectivity, which played a crucial role in regulating the energy budget of the earth-atmosphere system (Twomey, 1977). In addition,as the source of cloud condensation nuclei (CCN), SSA indirectly affects the formation of

cloud droplets, the optical properties of clouds, and the precipitation process, thereby altering the cloud radiative effect and precipitation distribution (Andreae and Rosenfeld, 2008). The SSA CCN number fraction observed in high latitudes in the Southern Hemisphere is up to 65 % (Quinn et al., 2017)

SSA primarily originates from the ocean surface (Stokes et al., 2013). Wind stress causes wave breaking, which entrains air bubbles into the sea surface. During the bubble bursting process, film droplets and jet

droplets are formed, serving as the primary source of SSA (Monahan and Muircheartaigh, 1980; Mcdonald et al., 1982; Vinoj et al., 2010). Therefore, a strong correlation exists between SSA concentration in the marine boundary layer and horizontal wind speed. As reported by Mcdonald et al. (1982), when wind speed increased from 3.4 to 10 m s⁻¹, the sea salt concentration increased by a factor of 7-10.While wind speed largely determines SSA production, the observed SSA concentrations in mid-

to-high latitude regions with prevailing westerlies are significantly lower than those measured in low-latitude trade wind zones. This phenomenon has been attributed to the influence of cyclone-induced vertical convergence transport, which reduces SSA concentrations in mid-latitude regions under comparable mean wind speeds (Shi et al., 2023).

Extratropical cyclones (ETCs) serve as critical components of atmospheric circulation, facilitating not

only transoceanic transport of aerosols from continental sources to the open ocean but also enabling their vertical redistribution through the troposphere (Humphries et al., 2016; Hu et al., 2022; Salim et al., 2023). Long-range advective transport by cyclones in the Polar Frontal Zone of the Southern Ocean (SO) accumulates anthropogenic aerosols within baroclinic boundaries, resulting in elevated dust and black carbon aerosol loading in this region (Salim et al., 2023).A previous study has demonstrated significantly

enhanced SSA aerosol optical depth (AOD) over oceans during cyclones compared to non-cyclonic periods. The Northern Hemisphere exhibits a mean enhancement of 130 %, with distinct seasonal and latitudinal variations: lower values during autumn/winter than spring/summer, and reduced AOD at higher latitudes relative to lower latitudes (Naud et al., 2016). Increased wind speeds driving higher sea salt emissions and elevated relative humidity inducing hygroscopic growth of aerosol particles account



for the observed AOD enhancement. The impact of cyclones on aerosols varies greatly across different oceans. Sea salt-like AOD contributes more uniformly to total AOD within Atlantic cyclones, whereas in Pacific systems it shows greater extension into the cyclone's cold sector (Sakerin et al., 2022).

The SO hosts one of the highest frequencies of cyclogenesis globally (Simmonds et al., 2003). Over recent decades, both the total number and intensity of SO cyclones have increased, culminating in October 2022 with the most intense extratropical cyclone observed in the satellite era (Lin et al., 2023). Studies reveal that convergent uplift by cyclones transports over 23.4 % of SSA to higher altitudes, creating surface SSA depletion zones (Shi et al., 2023). However, constrained by extreme environmental conditions and observational challenges, current research on ETCs impacts on aerosols remains predominantly focused on the Northern Hemisphere.

In this study, GEOS-Chem model with cruise observation was employed to understand the emissions characteristics of SSA during the cyclones in the SO. We used the composite analysis approach to investigate the generation, distribution, and transport of SSA during cyclone events by identifying and tracking extratropical cyclone trajectories. The findings address critical observational gaps and significantly advance understanding of the driving mechanisms governing sea salt aerosol production and transport in the SO.

## 2 Materials and methods

### 2.1 Cruise measurements

High-resolution observations of SSA composition concentrations were observed with R/V Xuelong from February 23 to April 8, 2018. The research domain spanned from the 0° S to 76° S and 148° E to 109° W, encompassing the majority of the Southern Ocean. To minimize contamination from ship exhaust, the sampling inlet was mounted on the bow mast 20 meters above sea level, positioned upwind of the vessel's smokestacks.

The online ion chromatography analyzer (URG-IGAC) and the single particle aerosol spectrometer (SPAMS) was employed to hourly analysis of water-soluble inorganic ions in aerosols. The URG-IGAC system comprises three core components: Wet Annular Denuder (WAD), Steam-jet Aerosol Collector (SJAC) with inertial impaction, and Ion Chromatography (IC). This configuration allows selective sampling of different aerosol size fractions (PM10, PM2.5) based on monitoring requirements.



SCI captures particulate matter in aerosols. Aerosols are drawn through the annular space between the inner and outer tubes of WAD at a flow rate of 16.67 L min$^{-1}$. The steam generation unit of the aerosol

processor directs steam nozzles toward the aerosol inlet, causing incoming aerosols to collide with steam. Coarse particles are immediately captured, while fine aerosols absorb moisture and pass through a condensation growth chamber where their mass and volume rapidly increase. These enlarged particles are then collected via inertial impaction. The captured aerosols form an aqueous solution state, which is analyzed by IC for dissolved cations and anions. The detection limit for Na$^+$ in aqueous solution is 0.03

µg L$^{-1}$.

### 2.2 GEOS-Chem model description

The GEOS-Chem model, a global 3D atmospheric chemistry model driven by assimilated meteorological data, has been widely validated and employed to assess regional and global aerosol process, including marine and polar aerosols (Tian et al., 2024; Huang and Jaeglé, 2017; Jaeglé et al., 2011). It incorporates

advanced modules for aerosol physical processes-including transport, deposition, and emissions-alongside critical emission inventories. Natural sea salt emissions in GEOS-Chem are calculated online based on meteorological fields. SSA source function integrates wind-speed-dependent parameterizations established by Gong (2003) and Jaeglé et al. (2011) further refines this through empirical regression to introduce sea surface temperature dependency parameters.

In this study, the GEOS-Chem (v12.6.0) model driven by MERRA-2 meteorological reanalysis data was implemented at 2° × 2.5° horizontal resolution with 47 vertical layers, simulating the global atmosphere from the surface to 0.01 hPa. Simulation was performed for the period of February 23 to April 8, 2018, after a 1-month spin-up period.

### 2.3 Identification of the Southern Hemisphere Extratropical Cyclones

Cyclone tracks were obtained from the dataset published by Mcerlich et al. (2023),which was generated using an automated tracking algorithm applied to ERA5 reanalysis data. This dataset includes extratropical cyclones south of 30° S during 2003–2019, with a minimum lifetime of 24 h and track length of 100 km. Only complete cyclones within the Southern Ocean sector (40° S–80° S, 140° E–110° W) that met the selection criteria were utilized in the analysis for the year 2018.



**2.4 Compositing Approach**

Many studies have employed cyclone-centered composite analysis (Jaeglé et al., 2017; Govekar et al., 2014; Robinson et al., 2023). While this approach obscures variability among individual cyclones, it effectively reveals universal characteristics of physical quantities and their spatial distributions associated with cyclone systems. The data within a longitudinal range of ±60° and latitudinal range of ±30° centered on the cyclone center were selected and resampled onto a regular grid of 4000 km × 4000 km with a spatial resolution of 100 km using the nearest-neighbor interpolation method. The composite field was generated by spatiotemporal averaging of all interpolated cyclone data. In this analysis, we focused on cyclones under sea ice-free conditions, thereby excluding potential interaction between sea-ice and aerosol generation.

To quantify anomalies relative to background conditions, 4,000 km × 4,000 km grid points were generated at the identical locations of the original cyclones for all time steps. The background field was then obtained by averaging over time. This approach separates the cyclone-specific contributions from the environmental conditions. Anomalies (e.g., ΔAOD) represent percentage enhancements during cyclone events relative to background values: $\Delta AOD = (AOD_{cyclone} - AOD_{background}) / AOD_{background} \times 100$.

**2.5 Supporting Data**

Hourly parameters including air temperature, relative humidity, and wind speed along the cruise track were obtained by the automated weather station equipped in the R/V Xuelong. SST measurements were sourced from Xuelong cruise. Sea ice concentration, mean sea level pressure, and precipitation data were obtained from the ERA5 reanalysis dataset (https://cds.climate.copernicus.eu/)

**3 Results and Discussion**

**3.1 SSA variations during the cyclone processes with SO cruise observation**

$Na^+$ is the key component of SSA which is commonly regarded as a tracer for SSA (Teinilä et al., 2014). The formation mechanism of SSA exhibits a multifactorial nature, influenced by factors such as wind speed, SST, sea ice concentration, and RH (Mcdonald et al., 1982; Prijith et al., 2014; Yan et al., 2020; Forestieri et al., 2018; Saliba et al., 2019). The variations in meteorological conditions during the cruise are shown in Fig.S1 in the Supplement. During the cruise, RH were consistently high, with periods below



70 % accounting for only 13.24 %. Consequently, the impact of RH on the hygroscopic growth of SSA was considered negligible. The mass concentration of $Na^+$ along the entire cruise track ranged from 14.42 to 7436.44 ng m$^{-3}$. Influenced by high wind speeds and elevated SST, the maximum concentration peaked

at 20–30°S. Sea ice near Antarctica restricts sea-air contact,resulting in low SSA concentrations. The correlation coefficient between wind speed and $Na^+$ concentration in mid-to-high latitude oceans was 0.33, lower than values reported in previous studies (Fig.S2). At the same time, less than the 980 hPa pressure level accounted for 46.11 % of observations. Low pressure often signifies the presence of cyclones (Naud et al., 2021), and cyclones feature strong vertical mixing of air currents. So, it suggested

that the observed low SSA concentrations were likely influenced by ETCs.

The research vessel was considered influenced by a cyclone when the distance to its center was less than the cyclone's radius. During the cruise, the vessel encountered eight cyclones. Except for Cyclone 3, all cyclones were associated with a decrease in atmospheric pressure, while rapid increases in wind speed were observed during all cyclone events. Precipitation during Cyclones 2, 6, and 8 promoted wet

deposition of SSA. Cyclones 4 and 5 occurred over sea ice-covered regions, Consequently, these cyclones were excluded from the analysis of cyclone impacts on SSA. Observations of the remaining cyclones revealed consistently lower SSA concentrations during all cyclonic periods compared to non-cyclonic periods. Cyclones are typically associated with strong winds that significantly enhance sea salt emission fluxes from the ocean surface (Weng et al., 2020). This result indicates that during cyclonic

events, SSA undergoes vertical transport to higher atmospheric levels via converging updrafts in the lower-level cyclone centers. Due to the wide spatial coverage of cyclones, ranging from hundreds to thousands of kilometers, and the fact that cyclones impact on SSA transport have not been quantified, we use the GEOS-Chem model for further analysis.





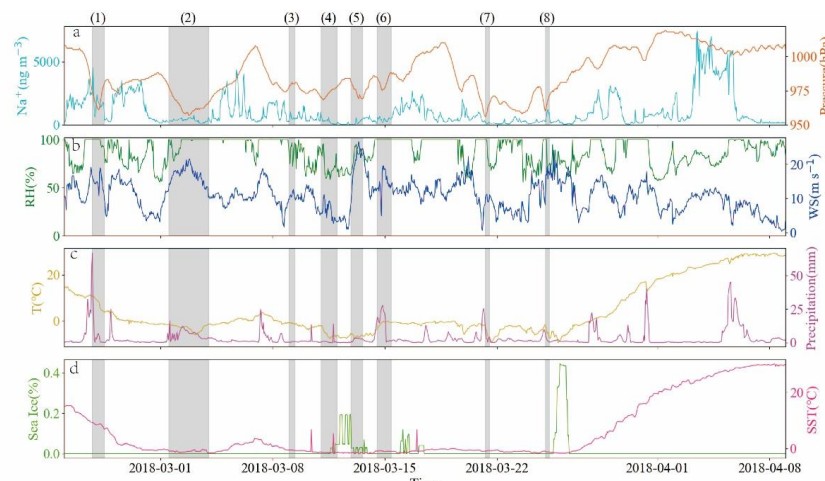

**Figure 1: Temporal distribution of Na+ and related meteorological parameters in the route of the 34th Chinese Antarctic scientific research vessel. (a) Time series of Na+ (light blue) and air pressure (orange). (b) Time series of relative humidity (dark green) and wind speed (blue). (c) Time series of air temperature (yellow) and precipitation (purple). (d) Time series of sea ice coverage (green) and sea surface temperature (magenta). Shading indicates cyclone periods.**

## 3.2 SSA spatial distribution during ETCs periods

The GEOS-Chem model enables analysis of aerosol physical processes including transport, deposition, and emission (Lin and Mcelroy, 2010; Wu et al., 2007; Wang et al., 2014). The simulation period spanned from February 23 to April 8, 2018. The simulated SSA concentrations exhibited a significant positive correlation (R=0.464) with observed Na+ concentrations (Fig.S3). After excluding sea ice-affected grid points in the model, the correlation coefficient increased to 0.472. Given the high complexity and observational uncertainty of atmospheric transport processes, the model demonstrates reasonable skill in capturing spatiotemporal SSA variations. The implemented parameterization scheme effectively characterizes SSA production and dispersion mechanisms.

AOD quantifies solar radiation attenuation by atmospheric aerosols at the earth's surface. Higher AOD values indicate greater aerosol concentrations and stronger light attenuation (Li et al., 2016; Mancinelli et al., 2024; Singh et al., 2024). Given that SSA was predominantly concentrated within the troposphere, atmospheric columns below 15 km formed the focus of the research. The composite analysis of the cyclone SSA AOD reveals that the spatial distribution characteristics of the SSA AOD are mainly presented as the high value area to the north of the cyclone center, which extends in a band along the northwestern direction, and the maximum AOD (0.136) occurs 500 km directly north of the cyclone



center (Fig.2a). Compared with the background environment, the AOD in the northwest quadrant of the cyclone increases up to 58.3 % (Fig.2b), while the southern region shows a decrease, which is opposite to the distribution characteristics in the northern hemisphere (Robinson et al., 2023).

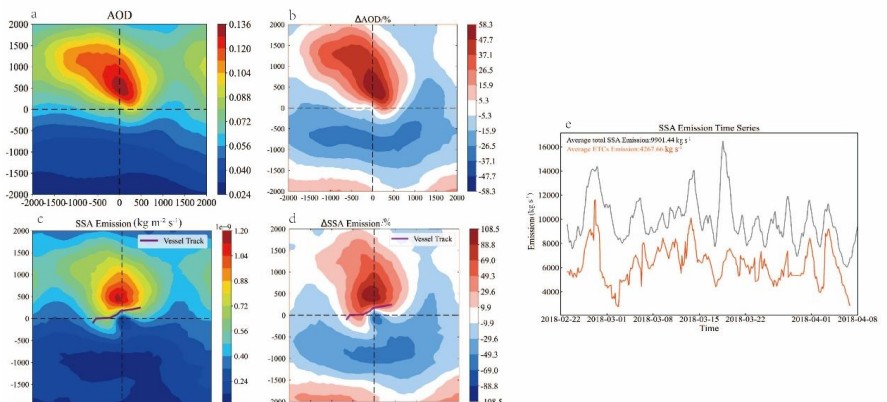

**Figure 2: Mean composites of SSA AOD and SSA emission in extratropical cyclones from GEOS-Chem model. (a) ETCs composite of SSA AOD. (b)△SSA AOD. (c) ETCs composite of SSA emissions (kg m$^{-2}$ s$^{-1}$). (d)△SSA Emission. (e) The SSA emissions from cyclones in high-wind-speed areas during the study period. The black line represents the emissions for the entire study area, while the red line indicates the emissions for the high-wind-speed regions.**

The high-value areas of AOD largely coincide with the regions of high SSA emissions, reaching a maximum of $1.20\times10^{-9}$ kg m$^{-2}$ s$^{-1}$ (Fig.2c), with the maximum increase amounting to 99 % (Fig.2d). There is a strong correlation between SSA emissions and wind speed and a study has shown that a 10 % increase in wind speed leads to a 38.4 % increase in calculated global emissions (Weng et al., 2020). The distribution of SSA emissions aligns with the distribution of 10-m wind speeds and 10-m wind speed distribution exhibits stronger values in the northern sector (>12 m s$^{-1}$ maximum) (Fig.3e), which is also found by Grandey et al. (2011). Research indicates that cyclone-induced high wind speeds significantly enhance the emission of SSA, leading to a substantial rise in regional SSA concentrations. Taking Cyclone 1 as an example, observational data shows that the SSA emission distribution during navigation is relatively uniform, with a slight downward trend in △SSA emissions. Accordingly, it can be inferred that the observed low SSA concentration is likely caused by the convergent updraft induced by the cyclonic circulation.

The total emissions of SSA in the high-wind zone (defined as the area within 2000 km of the cyclone center with wind speeds > 5 m s$^{-1}$) were used to assess the contribution of ETCs to sea salt aerosols. On



average, SO cyclones cause SSA emissions of 4016.798 kg s$^{-1}$, accounting for 63 % of emissions in the

study area (Fig.2e). Correspondingly, ETCs provide 60 % of sea salt emissions through strong wind

mechanisms over less than half of the ocean area in the Northern Hemisphere (Robinson et al., 2023).This

indicates that more than half of the SSA in the oceans originates under cyclonic conditions. In fact, in

regions with approximately 30 % sea ice coverage, the dynamic interaction between seawater and sea ice

can actually enhance the emission of SSA (Yan et al., 2020). Therefore, the contribution of cyclones to

SSA emissions may be higher.

The distribution of AOD is not only related to emission levels but also associated with various

meteorological conditions. The meteorological conditions of cyclones during the study period are shown

in the Fig.3. The average sea level pressure of the 922 cyclone points reaches the minimum of 977.5 hPa

(Fig.3a). Although SST promotes SSA production by reducing ocean kinematic viscosity and surface

tension, thereby enhancing bubble entrainment efficiency and ascent velocity, its effect on SSA

concentration remains minimal under low wind speeds (Saliba et al., 2019; Liu et al., 2021). SST exhibits

a gradual southward decrease with a maximum temperature of only 13°C (Fig.3b), demonstrating that

SST exerted virtually no impact on SSA generation during cyclonic periods. The maximum precipitation

(12 mm day$^{-1}$) is observed in the vicinity of the cyclone center, exhibiting a characteristic comma-shaped

distribution that curved northeastward (Fig.3c), accompanied by relative humidity levels exceeding 90 %

(Fig.3d). Under high humidity conditions, precipitation further enhances the wet scavenging efficiency

of sea salt particles, resulting in the occurrence of low SSA concentrations. Therefore, the spatial

heterogeneity in the distribution of SSA is also partially influenced by the wet scavenging removal

mediated by precipitation.





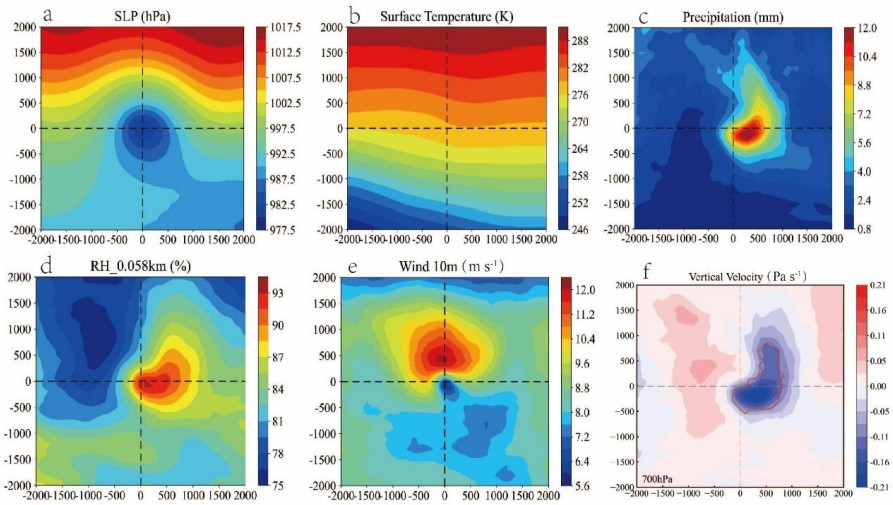


**Figure 3: The composites of meteorological conditions during ETCs period. (a) ETCs composite of Sea Level Pressure (hPa). (b) ETCs composite of Surface Temperature (K). (c) ETCs composite of SSA precipitation (mm). (d) ETCs composite of Relative Humidity (%). (e) ETCs composite of 10 m-wind (m s⁻¹). (f) ETCs composite of 700 hPa vertical velocity (Pa s⁻¹).**

**3.3 SSA vertical distribution and upward transport of ETCs periods**

Aerosols over the SO are predominantly concentrated in the near-surface mixed layer below 1 km. However, aerosol accumulation layers are also observed at specific altitudes due to atmospheric dynamic processes (such as updrafts and temperature inversions) (Alexander and Protat, 2019). The cyclone's meridional cross-section shows that SSA is primarily concentrated about 1,000 km to the north of the

cyclone, with its vertical distribution reaching a maximum height of 1.75 kilometers and decreases as altitude increases (Fig.4a). The concentration of SSA on the southern side of the cyclone is nearly zero. The high-value area of SSA on the northern side gradually shifts from the center toward the west, reaching its maximum displacement distance at an altitude of 1 km (Fig.4b).

The composite analysis of SSA mass concentration and number concentration show a trend of shifting

northwestward with increasing altitude (Fig.S4、Fig.S5). This vertical displacement pattern corresponds closely to observed variations in regions of high RH (Fig.S6). Vertically, the maximum SSA mass concentration decreases from 27 μg m⁻³ to 12.0 μg m⁻³ between the surface and upper levels. Concurrently, the peak anomaly increases from 57 % to 90 %. In contrast, SSA number concentration anomaly maintains relatively stable values throughout the atmospheric column. At 1 km altitude, collocated

maxima in both SSA number and mass concentrations are observed. This spatial covariation, combined



with the distribution of elevated RH, demonstrates that boundary layer airflow facilitates both the transport of sea salt aerosols and their moisture-driven hygroscopic growth.

The upward motion of ETCs is mainly dependent on the warm conveyor belt (WCB), which originates in the warm sector of the cyclone and rapidly ascends to the mid- and upper troposphere to redistribute the air mass (Browning and Roberts, 1994; Madonna et al., 2014). The boundary of the WCB was defined as the outermost closed contour of 700 hPa vertical velocity ($\omega$). This contour was required to be located within 1000 km of the cyclone center and to encompass the region of maximum ascent (the minimum $\omega$ value). The strongest upward motion associated with the WCB occurs at the cyclone center, coinciding with the location of the precipitation area—a configuration consistent with atmospheric dynamics (Fig.3f). For each cyclone, the average SSA vertical transport flux within the identified WCB region was calculated. The results demonstrate that cyclones alter the vertical transport direction of SSA, and the upward transport flux of SSA in the WCB reach its maximum value of $5.33 \times 10^{-11}$ kg m$^{-2}$ s$^{-1}$ at 450 m (Fig.4c). Surface friction suppresses the dynamic upward transport of SSA from the ocean surface, making it difficult for SSA generated at the sea surface to disperse to higher altitudes. At the same time, the cyclonic circulation causes the lower-level air to continuously converge toward the system center and undergo upward vertical motion. This dynamic process converges and transports surrounding SSA together to higher altitudes, ultimately resulting in a unique distribution pattern where the maximum concentration occurs 450 m rather than the surface.

The vertical transport of SSA by 26 complete cyclones vary from 51,218 kg to 2,610,570 kg, with the transport capacity jointly determined by the intensity and duration of the cyclones. Duration demonstrates a stronger correlation to total transport mass (R=0.51). Nevertheless, for cyclones of comparable duration, transport mass increases systematically with cyclone intensification (Fig.4d). The number of extreme cyclones is increasing at an average rate of 0.11 cyclones per year, with all cyclones exhibiting significant deepening of central pressure over the SO (Lin et al., 2023). This trend indicates that progressively more SSA would be transported, potentially amplifying important impacts on global climate systems.



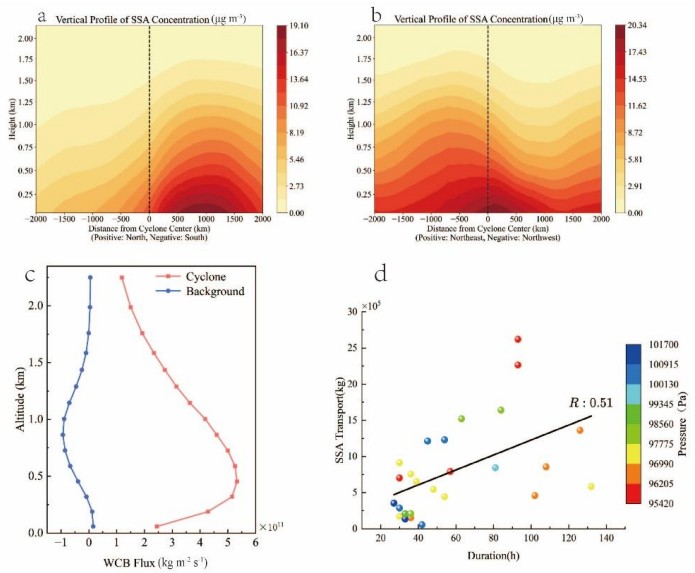

**Figure 4: Vertical profiles of SSA concentration and vertical flux. (a) North-south orientation of the cyclone center. (b) East-west orientation on the northern side of the cyclone center. (c)The mean vertical profile of SSA transport flux in the WCB of ETCs. (d)The relationship between the vertical transport of SSA and ETCs pressure as well as movement durations.**

### 3.4 The contributions of various physicochemical processes on SSA Budgets during ETCs

Figure 5a summarizes the contributions of various physicochemical processes to the tropospheric column mass budget of SSA within cyclone systems. The sum of diagnostic values for all grid cells within a 2000 km radius of the cyclone center was considered as the characteristic of SSA mass variation associated with ETCs. The emission process acts as the surface source term for SSA, transferring material from the surface to the atmosphere, while dry / wet deposition functions as the surface sink term, facilitating the transfer of atmospheric material to the surface. The cyclone-affected area is the strong source region of SSA, with emissions far exceeding deposition amounts.

Turbulent mixing process primarily refers to continuous, small-scale, non-directional vertical exchange aimed at reducing vertical gradients. This process drives the upward transport of 15,060 kg s$^{-1}$ of SSA mass from high-concentration regions near the surface to lower-concentration layers aloft. Convective processes, in contrast, specifically refer to rapid vertical mass transport triggered by deep and shallow convection mechanisms. Within the northern sectors of cyclonic systems, large-scale subsidence



dynamically compresses and forces upper-atmosphere air downward to the surface, resulting in net mass

loss of sea salt aerosol within the air column (Fig. S7).

Composite analysis of the advective transport process reveals that the northwestern quadrant of the cyclone center serves as the primary export region for SSA (2,026 kg s$^{-1}$), while the northeastern quadrant functions as the main import region (1,650 kg s$^{-1}$) (Fig. S7). This yields a net dissipative flux of -376 kg s$^{-1}$. Notably, the advective transport encompasses both the west-to-east transport of SSA and the upward

flux from the surface, both of which collectively influence the distribution of SSA. The turbulent mixing process plays a dominant role in the vertical transport of SSA.

Therefore, during ship-board observations, the relative spatial positioning between the research vessel and cyclones plays a crucial explanatory role in the occurrence of low SSA concentration phenomena (Fig.5b). Specifically, when navigating within high-wind regions near cyclone centers, the dominant

turbulent mixing process among various physical-chemical mechanisms significantly enhances vertical aerosol transport. This results in rapid lifting of substantial SSA particles to higher altitudes, leading to significantly reduced SSA concentrations near the sea surface. When positioned in low-wind areas south of cyclones, inadequate sea salt emission flux becomes the primary limiting factor preventing increases in SSA concentrations.



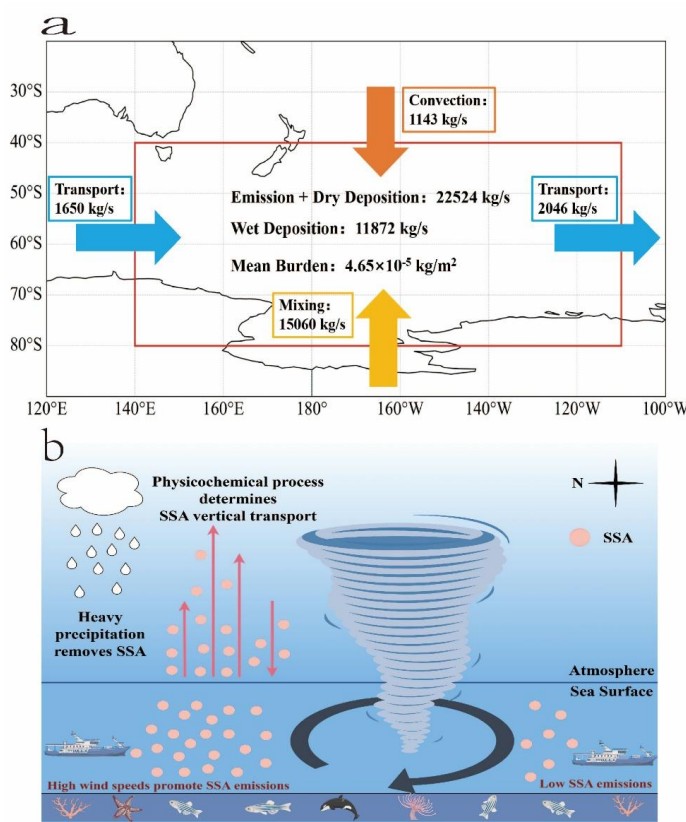

**Figure 5: The average budget of SSA during ETCs. (a) The yellow arrow represents transmission from lower levels to higher levels, and the orange arrow indicates transmission from lower levels to higher levels. (b) Interpret the small SSA concentration pattern in the schematic based on ship-cyclone positional relationships.**

## 4 Conclusions

Based on observational data and model simulations, the impacts of extratropical cyclones (ETCs) in the SO on the distribution and transport of sea salt aerosols (SSA) were systematically analyzed. Observational data from the 34th Antarctic expedition cruise reveals that SSA concentrations remained low during cyclone periods, which contradicts the expected positive correlation between wind speed and SSA concentration. GEOS-Chem model simulations were conducted to investigate this phenomenon.

The results demonstrate significant spatial heterogeneity in SSA distribution during cyclones, with concentrations predominantly concentrated north of the cyclone center. The maximum increase in aerosol optical depth (AOD) reached 58.3 %. With increasing altitude, high-concentration zones gradually shift



from the direct north of the cyclone center to the northwestern quadrant. SSA emissions in high-wind regions associated with extratropical cyclones accounts for 60 % of total emissions in the Southern Ocean.

Cyclones exhibit strong vertical mixing motion. A separate analysis of the ascending Warm Conveyor Belt (WCB) regions shows that the SSA transport flux peaks at 450 meters. The turbulent mixing process plays a dominant role in the vertical transport of SSA. The turbulent mixing process is the physicochemical process that has the greatest impact on SSA budgets during ETCs. Ultimately, the spatial heterogeneity of SSA distribution implies that the relative position between research vessels and

cyclones leads to different interpretations of low SSA concentrations, which are jointly influenced by emission and transport processes. Aerosols remain a significant source of uncertainty in future climate research. Further analyze is required to understand how cyclone-generated or -transported SSA regulates ocean-aerosol-cloud-climate interactions across different seasons and regions are needed.

**Coda and Data availability.** The data and data discussed in this paper are available from the following
website: https://doi.org/10.5281/zenodo.16875380 (Zhang, 2025).

**Author contributions.** X.Z. analyzed the results and wrote the paper. X.Z., and R.T., conducted the simulations and analysis. J.Y. conducted the observations, proposed the research ideas and wrote the paper. S.W., and S.Z., contributed considerably to the interpretation of the results. H.X., and Q.Z., reviewed and improved the draft. H.S., and X.S., conducted the observation and data processing.

**Competing interests.** The contact author has declared that none of the authors has any competing interests.

**Disclaimer.** Publisher's note: Copernicus Publications remains neutral with regard to jurisdictional claims in published maps and institutional affiliations.

**Acknowledgments.** The authors gratefully acknowledge Guangzhou Hexin Analytical Instrument
Company Limited for the SPAMS data analysis and on-board observation technical assistance, and Zhangjia Instrument Company Limited (Taiwan) for the IGAC technical assistance and data analysis. The authors gratefully acknowledge the "Cultivating Innovation Team" Program of Third Institute of Oceanography.

**Funding.** This was supported by the National Natural Science Foundation of China (grant numbers
42376038); the Scientific Research Foundation of Third Institute of Oceanography, MNR (grant numbers 2024012, 2019024); the Response and Feedback of the Southern Ocean to Climate Change (grant no.



RFSOCC2020-2025), and the Chinese Projects for Investigations and Assessments of the Arctic and Antarctic (grant no. CHINARE2017-2020).

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
