# Peer review of "Extratropical cyclones drive the spatial heterogeneity distribution of Sea Salt Aerosol (SSA) and vertical transport in the Southern Ocean"

_EGUsphere, 2025_

## Author Comment (AC1)

We appreciate the reviewers' support for publication and will address the suggested corrections and clarifications thoroughly in our revised manuscript. In this response document, reviewer comments are shown in black, and our responses are provided in blue.

**Reviewer 1**

This work investigates the role of cyclones in generating and regulating sea salt aerosol (SSA) in the Southern Ocean. First, measurements of SSA and meteorological conditions from a cruise are presented. Then, simulations with GEOS-Chem and composite analysis with a catalog of cyclones are used to understand the dynamics and drivers of SSA during cyclonic periods. A quantification of how much each underlying process contributes to the SSA budget over the SO is then derived.

SSA is a key species for climate and undergoing strong changes. At the same time, the dynamics of cyclones are also evolving over the Southern Ocean. The topic of this paper is therefore of interest to the atmospheric community.

General comments

As stated in the specific comments below, some of the arguments that are made are not well presented and/or relatively weak, and many details about the figures are missing to actually understand what they represent. Additionally, except for a comparison of SSA concentrations no link is made between model and observation, which would largely benefit the quality of the manuscript. At this stage, I think the paper would reach the same conclusions without including the measurements. Furthermore, there are times where too much detail is provided (e.g. the authors explain what AOD is, the description of the instrument measuring [SSA] is 12 lines long, part of the model analysis concludes that higher winds generate more SSA in the model while it is clearly expected since it is how it is parameterized…), while at the same time the reasoning leading to key results is not clearly explained which make them sound more like extrapolations sometimes. Overall, a major work of clarification needs to be done to make the arguments clearer, stronger and improve the quality of this paper for it to be fit for publication in ACP.

Response: Thank you for reviewing our research and providing your constructive feedback. We have carefully revised and rewritten the key argumentative sections of the paper to address potential logical gaps or ambiguities found in earlier versions. We also clarified and corrected inappropriate arguments, ensuring all major conclusions are supported by explicitly presented data and solid reasoning. The revised manuscript now features a more comprehensive model evaluation, including comparisons with existing literature. Additionally, we incorporated satellite AOD observations to address the spatiotemporal limitations of cruise measurements and contrasted these with model simulation results to validate the model's accuracy. We have removed or condensed overly simple or verbose descriptions, added more detailed captions and legends for figures and tables, and provided a thorough explanation of the data presentation process. In the following, we answer to the comments point by point.

**Specific comments**

L11: "strong positive correlation with wind speed" - you say line 147 that the correlation is 0.33, lower than in other studies. I would not call this a strong correlation.

Response: Thank you for your insightful comment. We fully agree with your point that a correlation coefficient of 0.33 does not meet the conventional criterion for a "strong correlation". To address this inconsistency, we have revised the first sentence of the Abstract from "strong positive correlation with wind speed" to " positive correlation with wind speed" to accurately reflect the statistical result reported in Line 11.

L80: I would not say "majority" of the SO. Looking at figure S1 you actually cover the SO between

170E and 109W, which is a significant portion but clearly not the majority.

Response: Thank you for your good suggestion. Our study region (spanning 140°E to 110°W), constitutes a significant sector of the Southern Ocean but does not encompass the majority of it. We apologize for the imprecise language and have revised it in the manuscript. (Line 88)

L106: Is a 2°x2.5° horizontal resolution okay for appropriately reproducing cyclones and/or observations from the ship cruise?

Response: Thanks for your comments. We chose the 2°x2.5° resolution because it is the finest horizontal resolution achievable for global simulations using our GEOS-Chem-v12.6.0 classic model. To our knowledge, many previous studies (Wainwright et al., 2012; Priestley and Catto, 2022;

Baker et al., 2024; Bourdin et al., 2024) have successfully utilized this resolution to simulate and investigate various atmospheric processes, including tropical and extratropical cyclones, yielding reasonable results.

The mean radius of extratropical cyclones in the dataset has an average value of 1000 kilometers, with the 99.9th percentile approximately 27,00 kilometers, exceeding the resolution distance in the

Southern Ocean (Fig. R1). In other words, the key dynamical structures that influence the atmospheric composition over our ship-track region remain well resolved at 2°×2.5°. Furthermore, our analysis focuses on large-scale patterns and the mean atmospheric response to cyclones, rather than mesoscale features that would require finer regional nested simulations.

For these reasons, we consider the 2°×2.5° resolution appropriate for reproducing the cyclone occurrence and the environmental background information along the cruise. We also acknowledge that some bias between the model grid-cell averages and the point-based ship observations is inevitable, given the coarser spatial resolution of the model grid. To address this issue, we have incorporated additional satellite datasets in the revised manuscript to provide higher-resolution observational constraints and to allow a more detailed comparison and assessment of the model performance. (Line 242-255, Fig.S8,S9,Table S1,S2)

[Figure]

Fig.R1. Cumulative frequency of occurrence of the maximum cyclone radii reached by each cyclone track. The solid (dashed) red line shows the mean (99.9th percentile) value of the distribution (Mcerlich et al., 2023).

L119: "The data" - which data? The cyclone tracks? The GEOS-Chem fields? Both?

Response: Thank you for your comment. The term "The data" specifically refers to GEOS-Chem model fields. To avoid ambiguity, we have revised the relevant sentence in the manuscript. (Line

161)

L125: "4,000 km × 4,000 km grid points were generated at the identical locations of the original cyclones for all time steps" - I do not understand this sentence and thus the methodology. Please rephrase/clarify.

Response: Thank you for your comment. Obtaining the background field involves two steps: first, perform a temporal average over the entire study period to generate a climatological mean field; second, with the cyclone center as the origin, extract a 4000 km × 4000 km window at the same location in the climatological mean field. This yields a one-to-one background window corresponding to each cyclone window, allowing us to isolate the cyclone-induced anomalies from the large-scale environmental state. We have revised the sentence in the manuscript to the clarified version above to eliminate confusion. (Line 165-168)

L147: "lower than values reported in previous studies" - which studies? missing reference.

Response: Thank you for your comment. To address the missing references, we have added specific citations for the previous studies that reported higher values.(Line 193)

L147–148: "At the same time, less than the 980 hPa pressure level accounted for 46.11 % of observations" - okay but according to Fig S1 as soon as the ship was south of 60°S the surface pressure was almost exclusively below 980hPa, which is related to higher latitudes having lower pressures on average, not necessarily to cyclone occurence. Low pressure **anomalies** would be a better indicator of the potential role of cyclones, not absolute low pressures.

Response: Thank you for your good suggestion. The pressure anomaly at each observation point was calculated by subtracting the temporal average pressure over the study period from the pressure value recorded at that point. By constructing a time series of pressure anomalies, we found that low- pressure anomalies accounted for 71.5% of the cruise (Fig. R2), which indicates frequent cyclone activity over the Southern Ocean. We have revised the manuscript to reflect this finding and included the figure in the supplement. (Line193-195、Fig.S3)

[Figure]

Fig.R2. Temporal distribution of pressure anomalies in the route of the 34th Chinese Antarctic scientific research vessel.

L151: "The research vessel was considered influenced by a cyclone when the distance to its center was less than the cyclone's radius" - as you go on to show in the next parts of the manuscript, depending on where you are with respect to the cyclone center (e.g. north vs south), the effects of an ETC on [SSA] differ. In this part you are inferring the influence of cyclones on observed [SSA]

on the ship only by looking at whether the ship is within reach of the cyclone so this is incomplete information. I think you would gain a lot in clarity and robustness by merging sections 3.1 and 3.2, analyzing Figure 2 first and then look at which quadrant of each cyclone the ship was located in for

Figure 1. Otherwise the link/motivation from section 3.1 to section 3.2 seems a bit artificial.

Response: Thank you for your comment. We agree that the "distance-based" criterion does not fully describe complex cyclone effects. Our preliminary analysis of cruise observations revealed that surface SSA concentrations did not increase with wind speed as expected during most cyclonic events. However, due to the spatiotemporal limitations of the cruise data, these initial findings were insufficient to fully elucidate the complex influence of cyclones on SSA spatial distribution.

Motivated by this unexpected result and the constraints of the observational data, we turned to the

GEOS-Chem model to conduct a more comprehensive analysis. To further enhance the robustness of our conclusions and mitigate the inherent spatiotemporal limitations of the ship-based observations, we have added additional analysis based on CALIPSO observations in the revised manuscript.

In the revised manuscript, we further analyzed the impact of cyclones on AOD using CALIPSO

satellite data. As shown in the composite analysis for the study period (Fig.R3), the northwest quadrant of the cyclone-centered domain exhibited an enhancement in AOD, with a mean increase of 0.02 to 0.03 corresponding to a relative contribution exceeding 20-30%. It should be noted that the number of valid satellite samples during this specific period was limited, with a maximum of only 111 per grid cell. To ensure statistical robustness, we performed the composite analysis for the entire year of 2018, which is based on a much larger dataset with at least 500 valid CALIPSO

samples per grid cell (Fig.R4). The results show an obvious AOD enhancement in the northwest quadrant, with an absolute increase of 0.01–0.03 and a relative contribution of more than 10–30%.

The spatial distribution of AOD enhancement observed by CALIPSO, particularly the obvious increase in the northwest quadrant, aligns well with our model simulation findings. This spatial pattern aligns well with our model simulations. The consistency between satellite observations and model results reinforces the finding of a cyclone-induced, non-uniform distribution of sea-salt AOD.

We have integrated this CALIPSO-based analysis between the cruise observations and the model simulations (Line 221- 232, Fig.2), thereby creating a more logical progression of the revised manuscript.

[Figure]

Fig.R3. Mean composites of AOD from CALIPSO observations during the study period (22 February–8 April 2018).

Left: absolute difference between cyclone and background AOD (AOD_cyclone − AOD_background). Right:

relative difference in percentage [(AOD_cyclone − AOD_background) / AOD_background × 100%]. Only grid cells with at least 60 valid CALIPSO samples are shown.

[Figure]

Fig.R4. Mean composites of AOD from CALIPSO observations during 2018. Left: absolute difference between cyclone and background AOD (AOD_cyclone − AOD_background). Right: relative difference in percentage

[(AOD_cyclone − AOD_background) / AOD_background × 100%]. Only grid cells with at least 500 valid CALIPSO

samples are shown.

L158–159: Figure 1 indicates that not all the cyclones you consider are indeed associated with higher wind speeds. Some do, but some do not (e.g. 3, 4 and 7 have average to low wind speed).

Therefore your argument line 159—161 on the updraft explaining the lower [Na+] is not very strong.

Furthermore, there is intense precipitation at the ship location either during the cyclones (2,6,8 as you mention but also 5 to some extent) or right before (1 and 7, which you do not mention).

Precipitation before the event can contribute to cleaning the air, and therefore concentrations take time to recover which can explain why they are lower. The only two remaining events are 3 and 4.

Cyclone 4 shows well below average wind speeds, and it seems that (3) is probably misplaced by

ERA5 since the ship data shows a dip in pressure about a day earlier than the gray shade in Figure

1. At the same time as the actual dip before (3) there is precipitation, so wet scavenging. Overall then it seems to me that in Figure 1 precipitation and wind speeds are sufficient to explain why [Na+]

is lower during cyclones without needing the argument about updraft. This reinforces my previous point that the structure of the manuscript should be changed.

Response: Thank you very much for your exceptionally thorough and insightful analysis. We acknowledge that it is premature to discuss updrafts at this stage, so we have removed this section from the manuscript. To better analyze Na⁺ concentration variations and their environmental influences, we have redrawn the time-series data from the Chinese 34th Antarctic Research Expedition voyage. As shown in Fig.R5, although all cyclonic events were accompanied by increased wind speeds, only Cyclones 5 and 8 were associated with elevated Na$^+$ concentrations (Fig.R6), which differs from the previous assumption, prompting our investigation into cyclonic effects. Most other cyclonic events featured precipitation during their duration. Precipitation's cleansing effect may reduce Na$^+$ concentrations. However, we also note that during Cyclone 4, where no precipitation occurred, Na$^+$ concentrations remained low. This phenomenon may be influenced by the distribution of cyclones, but constrained by the spatiotemporal limitations of the cruising data available. Consequently, we subsequently utilized CALIPSO satellite data and the GEOS-CHEM model to aid our analysis. In the revised manuscript, we have added discussions on precipitation and wind speed, while introducing satellite data to support our findings. (Line 197-213, Fig.1, Fig.S5 , Fig.S6)

[Figure]

Fig.R5. Temporal distribution of Na+ and related meteorological parameters in the route of the 34th Chinese Antarctic scientific research vessel. (a) Time series of Na$^+$ (light blue) and air pressure (orange). (b) Time series of relative humidity (dark green) and wind speed (blue). (c) Time series of air temperature (yellow) and precipitation (purple). (d) Time series of sea ice coverage (green) and sea surface temperature (magenta). Grey shading indicates cyclone periods, and hatched shading represents cyclone periods with no precipitation.

[Figure]

Fig.R6. Bar chart comparing the mass concentration of SSA during non-cyclonic periods and cyclonic periods.

L159—160: "This result indicates that during cyclonic events, SSA undergoes vertical transport to higher atmospheric levels via converging updrafts in the lower-level cyclone centers" - I do not think you have clearly showed this. At this point this is merely a hypothesis.

Response: Thank you for your comment. We removed this statement from the manuscript and explained that it might be the cyclone that caused the decrease in [Na+] concentrations.(Line 210)

L173—175: correlation only is not enough to characterize model skill, at least bias and RMSE should also be looked at. All of these numbers should ideally be compared to those from other GEOS-Chem studies on sea salt aerosol to actually conclude that the model performs well. In other words, is 0.46 actually a good enough correlation? Could you validate other variables from GEOS-Chem using the cruise measurements, e.g. wind speed and precipitation to see if the differences in Fig S3 could be partly explained by biases in the modeled meteorology?

Response: Thank you very much for your good suggestions. We acknowledge that correlation alone is insufficient to describe model performance. Following your recommendations, we have supplemented the bias, RMSE, and NMB analyses (Table R1). The model underestimated SSA by 22%, which may be due to underestimations of wind speed and precipitation. To further verify the model's performance, we compared it with other published literature (Table R2). The results show that our model's correlation coefficient falls within a reasonable range (0.44–0.89). It should be noted that to enable comparisons between the GEOS-Chem model and observations, the observed Na$^+$ mass concentration was converted to SSA mass concentration using a conversion factor of 3.256 (Riley and Chester, 1971). This conversion factor is derived from the mass ratio of Na$^+$ in seawater. Since the observed aerosol was PM$_{2.5}$, the ratio of accumulation mode SSA AOD to total SSA was calculated to convert the observations into a format comparable with model data. (Fig.R7)

To further evaluate the model's performance against observations, we compared simulated AOD with CALIPSO satellite measurements. As shown in Fig.R8, the model-simulated spatial distribution of AOD shows good agreement with CALIPSO observations, capturing the major spatial patterns and dominant AOD features, with high-value centers primarily located between 40°S and 60°S. Quantitatively, the model underestimates AOD relative to CALIPSO. The simulated domain-mean AOD at 550nm (0.05) is approximately 28% lower than the observational mean AOD at 532nm (0.07). This low bias could be attributed to factors such as underestimation of surface wind speeds in the model, wavelength dependence, smoothing effects from coarser resolution, and sampling biases from the limited high-quality CALIPSO orbital data. We have added relevant data and figures to the manuscript to discuss the model's performance. (Line242-255, Fig.S8, Fig.S9,Table S1、Table S2)

Table R1. Description of GEOS-CHEM model performance of met variables and SSA

| | Pressure (hPa) | SST (K) | Wind (m s$^{-1}$) | Precipitation (mm/day) | SSA (μg m$^{-3}$) |
|---|---|---|---|---|---|
| Correlation | 0.99 | 0.97 | 0.85 | 0.53 | 0.55 |
| Bias[1] | 4.12 | -1.73 | -2.38 | -0.15 | -2.18 |
| RMSE[2] | 4.68 | 3.09 | 3.42 | 7.80 | 16.96 |
| NMB[3] | 0.41% | -0.62% | -21.72% | -4.40% | -21.93% |

[1]The mean bias is defined as mean (model- observation).

[2]The mean Root Mean Square Error is defined as sqrt (mean ((observation - model)$^2$)).

[3] The normalized mean bias is defined as mean ((model- observation)/ observation) ×100%.

Table R2. Comparison of the GEOS-CHEM model of SSA mass concentration with observations

| Region | | Time | Observation (μg m$^{-3}$) | Model (μg m$^{-3}$) | Correlation coefficient | NMB[1] | References |
|---|---|---|---|---|---|---|---|
| PMEL | SALC | 1993-2008 | —— | —— | 0.71 | +33% | |

| | | | | | | |
|---|---|---|---|---|---|---|
| (80°N-70°S) SALA | | —— | —— | 0.52 | +6% | Jaeglé et al.(2011) |
| Neumayer (70.7°S, 8.3°W) | 2001-2007 | 0.99 | 0.37 | | -63% | Huang and Jaeglé. (2017) |
| Dumont d'Urville ((66.7°S, 140°E) | 2001–2008 | 1.23 | 0.74 | | -40% | |
| CHINARE09/10 (20°N-80°S) | 2009–2010 | 13.09 | 5.62 | 0.53 | -133% | Jiang et al.(2021) |
| CHINARE11/12 (20°N-80°S) | 2011–2012 | 8.71 | 4.93 | 0.87 | -77% | |
| CHINARE13/14 (20°N-80°S) | 2013–2014 | 6.78 | 3.98 | 0.44 | -70% | |
| CHINARE17/18 (20°N-80°S) | 2017–2018 | 14.39 | 5.61 | 0.89 | -157% | |
| 34 Antarctic expeditions (0°-74°S) | 2017–2018 | 3.37 ($PM_{2.5}$) | 7.77 | 0.55 | -22% | This study |

[1] The mean normalized bias is defined as mean ((model- observation)/ observation) ×100%.

[2] SALA represents the Accumulation mode SSA, SALC represents the Coarse mode SSA.

[Figure]

Fig.R7. Averaged accumulation mode SSA AOD fraction. SALA AOD Fraction refers to the Accumulation mode SSA AOD /Total SSA AOD.

[Figure]

Fig.R8. Comparison of average AOD from the CALIPSO satellite and model simulation research during the observation period. CALIPSO AOD@532nm data regridded to 2×2.5°, with mode AOD at 550nm.

L175—178: Visually it seems that GEOS-Chem struggles to reproduce observed [Na+] even more during cyclonic periods. I think this can be a major limitation to this study since you want to understand what happens to sea-salt during these specific periods. Furthermore, some analysis of whether GEOS-Chem/MERRA-2 reproduces the same cyclones as ERA5 is needed to support the validity of the results as you cross two different data sources.

Response: We thank the reviewer for the comment regarding the model's performance. As shown above, we have added a more comprehensive model evaluation in the revised manuscript, which includes a comparison of simulated AOD with CALIPSO satellite observations and a comparison with previously published results. Overall, the model captures the key spatial patterns and high AOD values in satellite retrievals (Fig. R3). Quantitatively, consistent with previous studies, the GEOS-Chem model tends to underestimate marine AOD by approximately 22% in our case (Table 2). This bias is likely attributable to factors such as underestimation of surface wind speeds, grid-cell averaging effects, and uncertainties in model parameterizations. We acknowledge this bias; however, given that the primary focus of our study is the spatial pattern of the cyclone-induced inhomogeneous distribution of aerosol and that this pattern was consistently found in both the simulations and the satellite composite analysis, the systematic underestimation is not expected to alter our main conclusions. The model evaluation analysis has been included in the revised manuscript to ensure transparency (Line242-255, Fig.S8, Fig.S9,Table S1、Table S2).

In terms of ERA5 and MERRA2 data, both of them are widely used, state-of-the-art atmospheric reanalysis products that are considered reliable references in climate research and have been extensively applied in synoptic-scale studies, including cyclone tracking (Hersbach et al., 2020;

Gelaro et al., 2017). The use of ERA5 for cyclone tracking and MERRA-2 for driving the GEOS-

Chem model was limited by data availability and model requirements. The cyclone trajectories over the Southern Hemisphere were obtained from a published dataset provided by(Mcerlich et al., 2023), which is based on ERA5 sea-level pressure fields. The GEOS-Chem model is conventionally driven by MERRA-2 or GEOS-FP meteorological analysis data. We employed MERRA-2 due to its validated performance in simulating global and regional atmospheric constituents (Jaeglé et al.,

2017; Tulger Kara and Elbir, 2025; Lan et al., 2023).

To assess the potential influence of using two reanalysis datasets in different components of our analysis, we compared the sea-level pressure (SLP) fields from ERA5 and MERRA-2—the key variable for cyclone identification and tracking—over the study period. The comparison (Fig. R9)

shows a high degree of agreement between the two products: the mean SLP spatial patterns are nearly identical, and the point-wise temporal correlation exceeds 0.96 across most of the study area.

The root-mean-square difference (RMSE) is generally below 1.5 hPa, which is negligible in the context of synoptic-scale cyclone systems. These results indicate that the discrepancies between

ERA5 and MERRA-2 are minimal and do not significantly influence the identification or characterization of cyclones in our study. A discussion of dataset comparison has been added to the revised manuscript and Supplementary (Line 145-149, Fig. S5).

[Figure]

Fig.R9. Comparison of sea surface pressure between Merra-2 and ERA5 meteorological fields.

Fig S3: what is the meaning of the blue shade? Could you overlay the cyclone periods on this plot?

Response: Thank you for helpful suggestion to improve figure clarity. We address your questions and revisions as follows: The blue shaded area in the figure represents the area covered by sea ice.

Based on the suggestions, we have revised the figure to overlay cyclone cycles, thereby visually illustrating the spatiotemporal relationships among cyclones, sea ice coverage, and SSA

concentrations (Fig. R10). We have incorporated the modifications into the supplementary materials.

(Fig.S8)

[Figure]

Fig.R10. Time Series of SSA mass concentrations from Observation and GEOS-CHEM. The blue areas are the sea ice coverage zone, and the shaded areas are the cyclone zone.

Figure 2: the compositing technique needs to be further explained. The way I understood it, since it is cyclone-centered, these maps do not correspond to an actual geographical region, rather a

"cyclone-space". But then you overlay the vessel track which makes me really confused as to what is represented in this figure. Also, there is no mention of how many cyclones are part of this composite. This is essential information for the reader to assess the statistical significance of this composite analysis. Overall, there is more explaining to be done here.

Response: Thank you for your comments. You are right that the composite maps are constructed using a cyclone-centered coordinate system rather than a fixed geographical coordinate system. The overlaid vessel track is transformed into the same cyclone-centered coordinate system. For each data point along the vessel's actual geographical track, we calculated its relative position from the center of the nearest cyclone during cyclone periods. However, since the composite analysis is not merely derived from Cyclone 1, we removed this part from the manuscript.

If the center of a cyclone is located within the study area, the cyclone is included in the study. Our analysis reveals that a total of 79 cyclone trajectories have been observed in this region. Excluding data points associated with sea ice cyclones, the remaining number of cyclone points amounts to

922. The composite analysis is performed on these 922 data points. (Line 152-156)

L199–202: it seems that you use only model data to reach this conclusion, so it is expected from the start since the modelled SSA emission is parameterized based on wind speed. This is not a result.

Response: We thank the reviewer for pointing this out. We agree that this conclusion is an expected outcome of the model's parameterization rather than a new finding. Accordingly, we have removed this discussion from the revised manuscript.

L203: "observational data shows that the SSA emission distribution". There is no mention of observed SSA emission in this paper so I do not know what this is referring to.

Response: Thank you for your comments. As indicated above, we have removed the section regarding Cyclone 1 from the manuscript.

L207–210: How frequent are ETCs in the study area at that period, i.e. how many grid cells are in a cyclone at each given time step? You say that ETCs contribute to 63% of SSA emissions, but if the ETCs cover 63% of the grid cells then you cannot say that they enhance emissions, maybe the emission per grid cell is the same with or without cyclone but cyclones happen to cover 63% of the area. On the other hand if the ETCs cover say 10% of the space-time domain but account for 63%

of the emitted mass, then you can say that they strongly enhance emissions. This frequency/coverage argument is really missing here to be able to conclude on emission enhancement.

Response: Thank you for your good suggestions. We agree that cyclones' "enhance" emissions are critically dependent on their spatiotemporal coverage in the study area, not just their contribution to the total SSA emission. The high-wind-speed areas of extratropical cyclones (HWSA) are defined as areas within a 2000 km radius of the cyclone center where wind speeds exceed 5 m s⁻¹. We calculated the percentage of HWSA areas relative to the total study area, with an average value of

51%. Concurrently, the percentage of SSA emissions under HWSA conditions relative to the total emissions in the study area averaged 63% (Fig. R11). This discrepancy between spatial coverage (51%) and emission contribution (63%) leads us to conclude that cyclones indeed enhance SSA

production. We have supplemented the figure in the manuscript. (Line284-288,Fig.3f)

[Figure]

Fig.R11. The proportional contributions of the high-wind-speed areas of extratropical cyclones to the study area and their corresponding SSA emissions to the study area.

L218: "The average sea level pressure of the 922 cyclone points reaches the minimum of 977.5 hPa"
- in your measurements, SLP at the ship location is consistently below 975 hPa during cyclones. What do you make of this modeled minimum value above 975hPa? Is the model adequately reproducing these cyclones?

Response: Thank you for this comment. We agree that the comparison between the observed SLP (<975 hPa) and the model's average (977.5 hPa) requires careful interpretation. It is important to clarify that these two values represent fundamentally different things: the ship recorded instantaneous, point-specific value during a limited number of cyclone encounters, whereas the modeled average minimum SLP of 977.5 hPa is derived from the composite analysis of MERRA-2 –driven data, representing a statistical mean across a large dataset of 922 identified cyclone points over a broader spatial and temporal domain. Therefore, the discrepancy between two distinct data types is not necessarily an indication that the model is inadequately reproducing the intensity of cyclones.

As shown above, the MERRA-2 reanalysis generally shows good consistency in reproducing the overall SLP patterns as well as other key meteorological fields relevant to our study (Fig.R9, Table R1). Moreover, the SLP fields from MERRA-2 and ERA5 also exhibit a high level of agreement (Fig.R3, Fig.R4). In the revised manuscript, we have added analyses to further demonstrate the model performance of meteorological fields (Lines 144-149, 221- 232, 249- 255, 262-263, Fig.S5, Fig.2).

Figure 4c: what is the meaning of a WCB flux in the "background" case. And what are these background cases? How is this WCB flux calculated?

Response: Thank you for your comments. The "background" performs a temporal average over the entire study period to generate a climatological mean field and extract the warm conveyor belts (WCB) data at the cyclone's same location in the climatological mean field. Under the background case, the WCB flux refers to the amount of upward transport of SSA at the original locations corresponding to extratropical cyclone WCBs in the climatological mean state. The background cases thus represent the climatological mean state of the atmosphere, allowing us to isolate the specific contribution of cyclonic WCB to SSA transport fluxes. The boundary of the WCB was defined as the outermost closed contour of 700 hPa vertical velocity ($\omega$). The SSA flux within WCB

was computed as: Flux = ∫(SSA concentration × vertical velocity) dz. We have added to the revised manuscript.(Line 340-343, Fig.5)

L271—272: I do not see this on the figure. For example I can see blue dots above yellow and red dots in the 40—60hr duration window.

Response: Thank you for your comments. We appreciate you pointing out the apparent inconsistency in the 40–60h duration window, where some blue dots (representing weaker cyclones) appear above yellow or red dots (representing stronger cyclones). Upon re-examining the data, we acknowledge that the limited sample size of observed cyclone tracks may affect the statistical robustness. We have clarified in the manuscript to make it more precise. (Line351-355)

L290: "This process drives the upward transport of 15,060 kg s-1 of SSA" - it is unclear how you calculated this number. For reproducibility of your results you should state more clearly what this number (and the others) corresponds to and how it was obtained, ideally including a formula for the calculation.

Response: Thanks for your good suggestions. The upward transport flux of SSA is a standard diagnostic variable directly output by the model, specifically the 'BudgetTransportTrop'. This variable is calculated internally by the model at each time step as the change in the total column mass of the tracer (SSA in this case) before and after the convective parameterization scheme is applied. The value of 15,060 kg s-1 presented in the manuscript is the time-mean and window-summed result of this diagnostic flux. We calculate the contribution of each physical process during the cyclone period by using this method.

We introduced all the variables corresponding to physical processes, as well as detailed calculation methods in the Supplement. (Line370-371, Text S2).

L296—298: what about the southern quadrants? The net dissipative flux accounts only for northern quadrants, why is that?

Response: Thanks for your comments. The southern quadrant exhibits a slight decrease and an increase that largely offset each other and are therefore negligible compared with the northern quadrants. In the revised manuscript, we have added a description of the southern-sector response

Figure 5a: I like the idea of this figure, but it is misleading since it only represents results/numbers for a 6-week-long simulation. This should be clearly stated somewhere, otherwise the reader might think this is an annual mean characteristic, whereas it is only valid for a particular season. If you really wanted to make such a figure, I think you should use sea salt concentration fields from e.g. MERRA-2 directly and combined with the ETC catalog for a whole year, which then would give you more general numbers.

Response: We thank the reviewer for this insightful comment. We agree that the original figure could be misinterpreted, and we have now clearly indicated in the revised manuscript that the results are based on a 6-week simulation (Fig.6a). Regarding the use of MERRA-2 aerosol fields, we appreciate the reviewer's intention to improve the generality of the analysis. We explored this option but found our process-oriented study relies on diagnostic fluxes (e.g., turbulent mixing, convection and transport) that are not available in MERRA-2. A key advantage of our GEOS-Chem simulation is its ability to output these process-level diagnostics, which are essential for mechanistically understanding cyclone impacts on sea-salt aerosol. For these reasons, we retain the process-based analysis from the simulation while clearly stating the limited time period. Extending the analysis to year-scale is an important direction for our future work.

Figures 4, S4, S5: the different scales of colorbars make it hard to compare the panels. You should harmonize the colorbars to have the same min and max values.

Response: Thanks for your suggestions. We have modified the pictures in the manuscript to ensure that the colorbars remain consistent. (Fig.4,S10,S11)

Several times the past tense is used where it should not, which can make the sentence confusing.

Response: Thanks for your comments. We have already clarified this in the manuscript.

**Reviewer 2**

The manuscript investigates the role of extra-tropical cyclones (ETCs) in generating and distributing sea salt aerosol (SSA) throughout the Southern Ocean troposphere. Results from a recent cruise are presented that show low SSA concentrations during high wind speeds experienced during ETCs. This phenomenon is explored utilising modelling work with GEOS-chem and a composite analysis. Model results and composite analysis are further utilised to understand the spatial, including vertical, distribution of SSA relative to ETC activity, and quantify the SSA budget over the SO. Given the significant and increasing influence of ETCs on this region, which exhibits a significant impact on global climate, and the large impact of SSA on global radiation over oceans, the topic of this paper is important and falls within the scope of ACP.

General Comments

In its current form, the paper requires major revisions to be suitable for publication in ACP. The authors make several key statements which aren't backed up by Figures that are presented in the manuscript. The data may be there, but how it is plotted doesn't really support the arguments made in the text, weaking the overall story.

The model results and observations aren't particularly well-linked to each other. The observations are utilised primarily, from what I can tell, as weak justification for the modelling work, and correlations between the observations and model aren't shown to be significant. The paper is heavily weighted towards interpretation of the model results, which are valuable in their own right, but the observations could be utilised better. Utilisation of satellite data may also be a way to strengthen the results, particularly since the authors are utilising AOD data from the model.

Response: Thank you for reviewing our research and for your constructive feedback. We fully agree that clear graphical representations are essential for supporting scientific conclusions. We reviewed the entire manuscript to ensure every key statement is directly supported by corresponding, intuitive graphical evidence. We redrew or modified multiple figures to enhance their clarity in serving the argument. We strengthened all figure titles and captions while providing more specific guidance to readers within the text.We incorporated statistical metrics such as root mean square error, bias, and correlation coefficients, comparing them with other literature to further validate the model's performance. Additionally, we introduced MODIS satellite remote sensing AOD data and added an analysis comparing simulated AOD with satellite-observed AOD. This not only strengthens the evaluation of model performance but also enhances the credibility of sea salt aerosol distribution during cyclones. In the following, we answer to the comments point by point.

Specific Comments

· The methods section is lacking significant detail in several areas

○ Please add a map showing an overview of the cruise and a general description of the cruise and its weather conditions, whether you interacted with the sea-ice region (some maps in your supplementary information would suggest you got close), and if so, how much time you spent there.

Response: Thank you for your good suggestion. High-resolution observations of sea salt aerosol (SSA) composition and concentrations were conducted aboard the research vessel (R/V) Xuelong from February 23 to April 8, 2018. The vessel departed from New Zealand, traveled southward through the westerly wind belt, entered the Southern Ocean, and then moved northward to the Western Pacific

Ocean, thus concluding the cruise.

The study area covered from 0°S to 76°S latitude and 148°E to 109°W longitude, encompassing a large part of the Southern Ocean. During the cruise, the vessel faced varying weather conditions typical of the region: south of 40°S, westerly winds dominated; in high-latitude areas (south of 60°S), conditions were persistently cloudy, with occasional light precipitation and frequent cyclonic activity disturbances.

Notably, the vessel interacted with sea ice regions during the cruise (Fig. R1). The total duration of sea ice interaction was 86 hours, mainly in the high-latitude

Southern Ocean (65°S–76°S), where the vessel traveled through areas with 10–80%

sea ice coverage. This sea ice interaction period accounts for about 15% of the total cruise duration, providing valuable observations of SSA characteristics in sea-ice- influenced marine environments. We have added this to the supplement.(Line 85-

88, Fig.S1)

[Figure]

Fig.R1. The 34th Antarctic Expedition route with sea ice. Black lines indicate the research vessel's route, the green dot marks the starting point, the red dot denotes the endpoint, and lighter blue indicates higher sea ice coverage.

o Please provide further detail into how you avoided platform exhaust in your sampling, as the sampling inlet won't always just be "upwind of the vessel's smokestacks".

Response: Thank you for your good comment. In this study, to minimize the impact of self-contamination of the vessel on the observation results, the air inlet connecting to the monitoring instruments was fixed at the bow of the R/V, approximately 20 meters above sea level. Note that the primary source of pollution is the chimney, which is positioned at the stern of the R/V and about 25 meters above the sea level. Hence, the vessel's pollution emissions were mainly downwind of the sampling inlet, especially when the vessel was running.

We used the onboard meteorological sensors to continuously record the relative wind direction in relation to the bow. The data is considered "clean air" only when the relative wind direction remains consistently within the sector range of 60° to 300°.(Line92-97)

o What was the temporal frequency of your sampling? Was it an integrated sample over the full hour, or just a grab sample of ~1 minute during each hour? In either case, please detail how you avoided sampling ship exhaust during this period.

Response: Thank you for your good comments. Sampling was continuous throughout the cruise, with data recorded as hourly integrated samples. Each data point represents the average SSA concentration over a full 60-minute period. The sampling system operates at a constant flow rate of 16.7 L min to ensure consistent sampling per hour. To minimize the impact of ship emissions, the sampling inlet connected to the monitoring instruments was fixed on a mast (20m above the sea surface) located at the bow of the research vessel. Note that the major pollutants came from the chimney, located at the stern of the R/V and about 25m above sea level. Hence, the vessel's pollution emissions were mainly downwind of the sampling inlet, especially when the vessel was running forward. We have supplemented and clarified in the manuscript. (Line92-97)

o Further details about the method are required, whether this be a reference to another publication with further details, or adding all the additional details here. Also need to add the manufacturer and model IDs.

Response: Thank you for your good suggestion. Online measurements of sea salt aerosol (SSA) composition were conducted using an online ion chromatography analyzer (IGAC, Model S-611) and a single-particle aerosol spectrometer (SPAMS, Hexin Analytical Instrument Co., Ltd., China). The detection methods and operational procedures have been described in detail in previous studies (Yan et al., 2019; Li et al., 2014). The online ion chromatography analyzer, model S-611, is an advanced equipment. The ion chromatography (IC) system is manufactured by Dionex, now part of Thermo Fisher Scientific, with the specific model designation ICS-3000.We have supplemented the required information on the analytical method, including relevant references and details of the instrument manufacturer/model. (Line99-100,106)

o How was the detection limit for the IC determined? Can you describe the calibration undertaken for the method?

Response: Thank you for your insightful question regarding the determination of the detection limit and the calibration procedure for the ion chromatography system.

We prepared 6–8 standard solutions of $Na^+$ at gradient concentrations using a certified primary standard. All solutions were prepared with ultrapure water (18.2

$MΩ·cm^{-1}$) to avoid background contamination. The standard solutions were analyzed in triplicate using the Dionex ICS-3000 IC system under the same operational conditions as the sample measurements. A linear regression model was fitted between the measured peak areas of $Na^+$ and the corresponding standard concentrations. The calibration curve showed excellent linearity (R2 = 0.998), confirming the reliability of the quantification method over the tested concentration range.

The detection limit of the IC system for $Na^+$ was determined using the signal-to- noise (S/N) method, which is widely accepted in ion chromatography analysis (Xu et al., 2000; Pohl et al., 1999). We analyzed replicate blank samples under identical operational conditions to those used for the samples, with the aim of measuring background noise (N). This background noise is defined as the standard deviation of the blank peak areas, providing a crucial indicator of signal variability in the absence of any specific analyte. The detection limit was calculated as DL = 3 ×

(S/N), resulting in a DL of 0.03 $μg·L^{-1}$ for $Na^+$.

We have added these detailed calibration and detection limit determination procedures to the supplement. (Line106-109,Text S2)

• Line 123 – You excluded cyclones that went over the sea ice region. I would assume that given your sampling location, that many of your samples would have been obtained during cyclones that went over sea ice. You need to show that this choice of focussing on sea-ice- free cyclones is valid and applicable to your dataset.

Response: Thank you for your critical comments. The core objective of our study is to investigate the interaction between SSA and cyclonic activity. The presence of sea ice can significantly modify the interaction between the sea and the atmosphere. In particular, the extent of sea ice coverage plays a crucial role in the formation of sea salt aerosols (Yan et al., 2020; Bergner et al., 2025; Lai et al., 2025; Ranjithkumar et al., 2025). The presence and melting of sea ice also influence atmospheric stability, friction velocity, and heat exchange (Deng and Dai, 2022; Kawaguchi et al., 2024; Braithwaite, 1995; Mcphee, 1983).

Our GEOS-Chem model's SSA emission scheme is parameterized for open-ocean conditions and does not fully account for sea-ice-related aerosol sources.

During the observation period, a total of 91 cyclones were identified, of which 12 passed over sea ice, and 79 cyclones operated entirely in ice-free areas. Ice-free track points accounted for 95% of the total. This data indicates that the "sea ice-free cyclone" observation events we have collected are abundant, sufficient to support our analysis of the impact of cyclones on open-ocean surfaces. (Line 150-156)

However, we also recognize and need to elaborate on the possible limitations that may arise from this in the text. It must be noted that the conclusions drawn from this study are mainly applicable to cyclones that develop over open ocean surfaces. For cyclones that interact with sea ice, their aerosol-generating mechanisms may differ significantly, which is undoubtedly a topic worthy of in-depth exploration in future research. In subsequent research, we will further explore the aerosol generation mechanism in cyclones in the presence of sea ice, aiming to gain a more comprehensive understanding of the impact of cyclones on the ocean-atmosphere interaction under different environmental conditions.

- Line 128 – you suddenly introduce AOD as a measurement – is this a measurement utilised in your analysis or a model output? If so, please introduce it earlier in the text

  Response: Thank you for your comment. You are correct that the introduction of AOD was abrupt. We define the anomaly as $\Delta V = [(V_{cyclone} - V_{background}) / V_{background}] \times 100$, where $V_{cyclone}$ and $V_{background}$ represent the aerosol variables under cyclone conditions and background conditions, respectively. We have revised the manuscript. (Line168-173)

- Paragraph starting Line 151 – this paragraph needs to reference Figure 1 early in the paragraph, otherwise the reader is just taking the author's word for it until they discover the Figure.

  Response: Thank you for your comment. We have revised the text to explicitly cite Figure 1 at the beginning of the relevant section, allowing readers to contextualize the description with visual support from the outset. (Line197-198)

- Line 156 - stating that SSA is lower during cyclonic periods compared to non-cyclonic periods is an important statement that needs some summary data shown to back it up, since it is a central basis for the study.

Response: Thank you for your good suggestion. When we compared the eight cyclonic events encountered during the navigation period with the non-cyclonic period, we found that, contrary to expectations, only Cyclones 5 and 8 showed increased SSA concentrations. The remaining cyclones showed varying degrees of $Na^+$ concentration reduction, despite the increase in wind speed (Fig. R2). During the six cyclonic processes, the average $Na^+$ concentration was 580.60ng $m^{-3}$, which was lower than 900.10 ng $m^{-3}$ in the non-cyclonic period. We have supplemented this information in the manuscript. (Line 199-205)

[Figure]

Fig.R2. Bar chart comparing the mass concentration of SSA during non-cyclonic periods and cyclonic periods.

- Line 174 – I'm not sure I would classify a correlation coefficient of 0.464 as a "significant correlation". Figure S3 is just a time series of the results, and not a correlation plot. It might help you doing a correlation plot of all the data, together with the subset of data that you're actually comparing with the ETCs in addition to the time series.

Response: Thank you for your helpful suggestion. We agree that describing a correlation coefficient of 0.464 as "significant" may overstate its strength. In order to provide a more comprehensive evaluation of the model's performance, the correlation coefficient analysis was supplemented with additional statistical metrics, including bias, root mean square error (RMSE), and normalized mean bias (NMB) (Table R1). Furthermore, a scatter plot of SSA between the model and observation was plotted (Fig. R3). The results of these supplemental analyses reveal that the model tends to underestimate the concentration of SSA by approximately 22%. We compared the correlation coefficient within the context of existing GEOS-Chem studies and found that the model's correlation falls within the normal range (Table R2). It should be noted that to enable comparisons between the GEOS-Chem model and observations, the observed $Na^+$ mass concentration was converted to SSA mass concentration using a conversion factor of 3.256 (Riley and Chester, 1971). This conversion factor is derived from the mass ratio of $Na^+$ in seawater. Since the observed aerosol was $PM_{2.5}$, the ratio of accumulation mode SSA AOD to total SSA was calculated to convert the observations into a format comparable with model data. (Fig.R4)

To further evaluate the model's performance against observations, we compared simulated AOD with CALIPSO satellite measurements. As shown in Fig.R5, the model-simulated spatial distribution of AOD shows good agreement with CALIPSO observations, capturing the major spatial patterns and dominant AOD features, with high-value centers primarily located between 40°S and 60°S. Quantitatively, the model underestimates AOD relative to CALIPSO. The simulated domain-mean AOD at 550nm (0.05) is approximately 28% lower than the observational mean AOD at 532nm (0.07). This low bias could be attributed to factors such as underestimation of surface wind speeds in the model, wavelength dependence, smoothing effects from coarser resolution, and sampling biases from the limited high-quality CALIPSO orbital data. Based on the above phenomena, we believe the model can reproduce the SSA distribution, and we have incorporated this section into the manuscript (Line 242-255, Fig.S8, Fig.S9,Table S1, S2).

Table R1. Description of GEOS-CHEM model performance of met variables and SSA

| | Pressure (hPa) | SST (K) | Wind (m s$^{-1}$) | Precipitation (mm/day) | SSA (μg m$^{-3}$) |
|---|---|---|---|---|---|
| Correlation | 0.99 | 0.97 | 0.85 | 0.53 | 0.55 |
| Bias[1] | 4.12 | -1.73 | -2.38 | -0.15 | -2.18 |
| RMSE[2] | 4.68 | 3.09 | 3.42 | 7.80 | 16.96 |
| NMB[3] | 0.41% | -0.62% | -21.72% | -4.40% | -21.93% |

[1]The mean bias is defined as mean (model- observation).

654 [2]The mean Root Mean Square Error is defined as sqrt (mean ((observation - model)$^2$)).

655 [3] The normalized mean bias is defined as mean ((model- observation)/ observation) ×100%.

657    Table R2. Comparison of the GEOS-CHEM model of SSA mass concentration with observations

| Region | | Time | Observation (µg m$^{-3}$) | Model (µg m$^{-3}$) | Correlation coefficient | NMB[1] | References |
|---|---|---|---|---|---|---|---|
| PMEL (80°N-70°S) | SALC[2] | 1993- 2008 | —— | —— | 0.71 | +33% | Jaeglé et al.(2011) |
| | SALA[2] | | —— | —— | 0.52 | +6% | |
| Neumayer (70.7° S, 8.3° W) | | 2001-2007 | 0.99 | 0.37 | | -63% | Huang and Jaeglé. (2017) |
| Dumont d'Urville ((66.7° S, 140° E) | | 2001–2008 | 1.23 | 0.74 | | -40% | |
| CHINARE09/10 (20°N-80°S) | | 2009–2010 | 13.09 | 5.62 | 0.53 | - 133% | Jiang et al.(2021) |
| CHINARE11/12 (20°N-80°S) | | 2011–2012 | 8.71 | 4.93 | 0.87 | -77% | |
| CHINARE13/14 (20°N-80°S) | | 2013–2014 | 6.78 | 3.98 | 0.44 | -70% | |
| CHINARE17/18 (20°N-80°S) | | 2017–2018 | 14.39 | 5.61 | 0.89 | - 157% | |
| 34 Antarctic expeditions (0°-74°S) | | 2017–2018 | 3.37 (PM$_{2.5}$) | 7.77 | 0.55 | -22% | This study |

658 [1] The mean normalized bias is defined as mean ((model- observation)/ observation) ×100%.

659 [2] SALA represents the Accumulation mode SSA, SALC represents the Coarse mode SSA.

[Figure]

Fig.R3. Scatter plot of SSA between model and observation. Red scatters indicate cyclonic periods, while scatters indicate non-cyclonic periods.

[Figure]

Fig.R4. Averaged accumulation mode SSA AOD fraction. SALA AOD Fraction refers to the Accumulation mode

SSA AOD /Total SSA AOD.

[Figure]

Fig.R5 Comparison of average AOD from the CALIPSO satellite and model simulation research during the observation period. CALIPSO AOD@532nm data regridded to 2×2.5°, with mode AOD at 550nm.

• Line 196 and Figures 2b and d – I'm struggling to understand what your % is calculated relative to, and thus getting any meaning to it. Please clarify.

Response: Thank you for your comment. The percentage indicates the anomaly of the SSA

emission flux during cyclonic periods compared to the climatological mean SSA emission flux. The formula is as follows: $\Delta$SSA Emission = [(SSA Emission $_{cyclone}$ − SSA Emission

$_{background}$) / SSA Emission $_{background}$] × 100, where SSA Emission $_{cyclone}$ and SSA Emission

$_{background}$ represent the values under cyclone conditions and background conditions, respectively. We have clarified this in the manuscript. (Line 269-270,Fig.3)

• Line 209 – You state here that SSA emissions from ETCs are 4016.798 kg/s, but Figure 2e states they are 4267.66 kg/s. Figure 2e also suggests that the emissions from ETCs account for 43%, not 63% of total emissions, as is suggested in the text. Maybe Figure 2e as a timeseries isn't the best way to represent these data and communicate your point.

Response: Thank you for your comment. The inconsistencies occurred because an incorrect version of Figure 2e was mistakenly included in the original submission. This has now been corrected. The SSA emission rate from ETCs (6286.01 kg/s) matches the value shown in the corrected figure. The contribution of ETCs to total emissions (63%) is accurately derived from and reflected in the corrected figure. We again thank the reviewer for catching this error.

Meanwhile, we supplemented spatial coverage in high-wind-speed areas (Fig.R6), and the results showed that the high-wind-speed areas of extratropical cyclones are defined as areas within a 2000 km radius of the cyclone center where wind speeds exceed 5 m s⁻¹, which occupy an average of 51% of the total study area. Nevertheless, these zones accounted for

63% of total sea-salt aerosol emissions. This disproportion between spatial coverage (51%)

and emission contribution (63%) leads us to conclude that cyclones indeed enhance SSA

production. We have revised and added to the manuscript. (Line 281-288, Fig.3f)

[Figure]

Fig.R6. The proportional contributions of the high-wind-speed areas of extratropical cyclones to the study area and their corresponding SSA emissions to the study area.

.

•    Figures 3 and 4 are referred to quite a lot in the text, but not necessarily in order. Please reconsider the order in which you present the subfigures and how you split them between figures 3 and 4 such that they flow better with your text.

Response: Thank you for your excellent suggestion regarding the flow and presentation of

Figures 3 and 4. We have adjusted the position of Figure 3f and moved it to Figure 4c. The new Figure 3 (The new picture of the revised manuscript is Fig.4) focuses on elucidating the mechanism by which meteorological conditions influence the heterogeneity in the spatial distribution of SSA during cyclone periods. The new Figure 4 (The revised manuscript contains Fig. 5) primarily demonstrates the unevenness of SSA transport upward during the cyclone process, as well as the specific positioning of the warm conveyor belt and the SSA flux transported upward. At the same time, we also rearranged the order of each image in the group and made changes to the main text. (Fig.4, 5)

- Line 269 – where are the 26 cyclones coming from here? Your initial analysis was looking at less than half of that.

  Response: Thank you for your comment. The 26 cyclones refer to complete extratropical cyclones (ETCs) that formed, developed, and dissipated entirely within the study domain (40° S-80° S, 140° E-110° W) during the cruise period (February 23-April 8, 2018), with no sea ice encounters. This represents the full inventory of cyclonic activity in the study area during the observation period, regardless of whether the research vessel directly encountered the cyclones. The "fewer than half" cyclones mentioned in the initial analysis refer to vessel-observed cyclones. These are the cyclones that came near the vessel's path and directly affected the vessel's sampling environment, such as causing significant fluctuations in pressure and wind. We have clarified the definition of the 26 cyclones in the manuscript.(Line 351, 152-156)

Technical corrections

- Abstract – grammar and word choice needs to be reviewed and improved significantly to enable better communication of the results.

  Response: Thank you for your comment. We have improved the accuracy of grammar and word choice.(Line10-29)

- Line 29
  - "ocean boundary layer" should be "marine boundary layer"

    Thank you for your detailed identification of this language mistake. We have updated the expression in the manuscript. (Line 34)

  - By "largest aerosol" do you mean it has the largest diameter, or has the highest mass concentrations? Please clarify to make it consistent with the rest of the sentence.

Response: Thank you for your comment. The phrase "largest aerosol" specifically refers to the component with the highest mass concentration. This clarification has been added to the manuscript. (Line34-35)

- Line 30 – you mention here that the aerosol content is minimal, directly after you've stated it is the largest contribution to aerosol in the global ocean boundary layer. This seems contradictory. Please clarify.

Response: Thank you for your comment. We have clarified in the manuscript. (Line 35)

- Line 31 – SSA mass concentration doesn't reduce effective radius – aerosol number concentration does.

Response: Thank you for your comment. We apologize for the earlier misstatement and have carefully revised the relevant content to match basic aerosol physics. (Line 36)

- Line 34 – SSA is "a" source, not "the" source of CCN.

Response: Thank you for your detailed identification of this language mistake. We have revised the expression in the manuscript. (Line 38)

- Line 60 – the sentence ending in "…AOD enhancement" requires a reference.

Response: Thank you for reminding us to strengthen the statement with supporting references. We have added a relevant, peer-reviewed citation to the sentence to validate the link between cyclonic activity and AOD enhancement. (Line 65)

- Line 70 should read "In this study, GEOS-Chem model simulations, together with cruise observations, were employed…"

Response: Thank you for your detailed identification of this language error. We have corrected the expression in the manuscript. (Line 76)

- Line 80 – this is not "the majority" of the Southern Ocean. Please rephrase.

Response: Thank you for your comment. We have revised the text to use "significant portion" rather than "the majority". (Line 88)

- Line 88 – what is "SCI"? Please define.

  Response: Thank you for your question about the abbreviation "SCI". To enhance the conciseness of the manuscript, this content has been relocated to the supplement. We have updated the appendix to clearly define "SCI" upon its first appearance there. (Text S1)

- Please define SST and RH when you first introduce the acronyms.

  Response: Thank you for your helpful reminder to define acronyms for clarity. We have revised the manuscript to explicitly define "SST" and "RH" upon their first use. (Line 175, 177)

- Line 146 -please define what you latitudes you are using for "mid-to-high latitude oceans". Based on Figure S2, it looks to be 40-60oS, but it would be good to just include this in the main text for completeness.

  Response: Thank you for your careful observation and suggestion to clarify the latitude range for "mid-to-high latitude oceans." We have updated the main text to explicitly specify this latitude range for clarity. (Line 192)

- Line 180 – this is a very generic statement about AOD and I'd suggest these references therefore aren't suitable.

  Response: Thank you for your comment. We have updated the references to better match the content. (Line 221-222)

- Line 182 – please start this sentence with reference to the figure, to guide the reader better. For example, "As shown in Figure 2, the composite analysis…"

  Response: Thank you for your helpful suggestion to improve readability by linking the sentence to the figure at the beginning. We have revised the sentence to start with a clear reference to the figure. (Line 222-223)

- The quality of the figures needs to be improved, text is often too small and pixelation is occurring at the present resolution, making it difficult to read.

Response: Thank you for your valuable feedback regarding figure quality. We have taken comprehensive steps to fix these problems. The revised figures have replaced the original versions in the manuscript.

•  Figure 2e caption – the "red line" comes out as orange or brown in the figure. Also, what are you defining here as the "high-wind-speed regions"? Please quantify

Response: Thank you for your comments. We have addressed both points with the following revisions: The line previously described as "red" has been explicitly changed to orange in both the figure and the text to match its actual appearance in the visualization.

High-wind-speed regions are defined as the area within 2000 km of the cyclone center with wind speeds $> 5$ m s$^{-1}$. We have made corrections and supplements to the manuscript.

(Line282-283,Fig.3e)

•  Again, please introduce Figure 3 generally before jumping into discussing it.

Response: Thank you for your good suggestion. We have made revisions in the manuscript.

(Line 295)

•  Line 201 – "Research indicates…" – what research? Please add citations or rephrase.

Response: Thank you for your comment. Since SSA emissions are the result of a wind speed parameterization scheme, we have removed the relevant expressions from the manuscript.

•  Line 203 – please refer to some shown data in a Figure or something when you're "taking

Cyclone 1 as an example".

Response: Thank you for your suggestion. Since the synthetic analysis represents an average distribution rather than one specific to Cyclone 1, we have removed the analysis pertaining to Cyclone 1 from the manuscript to avoid any misunderstanding.

•  Figure 4c x-axis label – would this be more accurate as "WCB SSA Flux"?

Response: Thank you for your suggestion. We have revised the label.(Fig.5)

References

Baker, A. J., Vannière, B., and Vidale, P. L.: On the Realism of Tropical Cyclone Intensification in
Global Storm-Resolving Climate Models, 51, e2024GL109841, https://doi.org/10.1029/2024GL109841,
2024.

Bergner, N., Heutte, B., Beck, I., Pernov, J. B., Arnold, S. R., Angot, H., Boyer, M., Creamean, J.
M., Engelmann, R., Frey, M. M., Gong, X. D., Henning, S., James, T., Jokinen, T., Jozef, G., Kulmala,
M., Laurila, T., Lonardi, M., Macfarlane, A. R., Matrosov, S. Y., Mirrielees, J. A., Petäjä, T., Pratt, K. A.,
Quelever, L. L. J., Schneebeli, M., Uin, J., Wang, J., and Schmale, J.: Characteristics and effects of
aerosols during blowing snow events in the central Arctic, Elementa-Science of the Anthropocene, 13,
10.1525/elementa.2024.00047, 2025.

Bourdin, S., Fromang, S., Caubel, A., Ghattas, J., Meurdesoif, Y., and Dubos, T.: Tropical cyclones
in global high-resolution simulations using the IPSL model, Climate Dynamics, 62, 4343-4368,
10.1007/s00382-024-07138-w, 2024.

Braithwaite, R. J.: Aerodynamic stability and turbulent sensible-heat flux over a melting ice surface,
the Greenland ice sheet, Journal of Glaciology, 41, 562-571, 10.3189/S0022143000034882, 1995.

Deng, J. C. and Dai, A. G.: Sea ice-air interactions amplify multidecadal variability in the North
Atlantic and Arctic region, Nature Communications, 13, 10.1038/s41467-022-29810-7, 2022.

Gelaro, R., McCarty, W., Suárez, M. J., Todling, R., Molod, A., Takacs, L., Randles, C. A.,
Darmenov, A., Bosilovich, M. G., Reichle, R., Wargan, K., Coy, L., Cullather, R., Draper, C., Akella, S.,
Buchard, V., Conaty, A., da Silva, A. M., Gu, W., Kim, G.-K., Koster, R., Lucchesi, R., Merkova, D.,
Nielsen, J. E., Partyka, G., Pawson, S., Putman, W., Rienecker, M., Schubert, S. D., Sienkiewicz, M., and
Zhao, B.: The Modern-Era Retrospective Analysis for Research and Applications, Version 2 (MERRA-
2) %J Journal of Climate, 30, 5419-5454, https://doi.org/10.1175/JCLI-D-16-0758.1, 2017.

Hersbach, H., Bell, B., Berrisford, P., Hirahara, S., Horányi, A., Muñoz-Sabater, J., Nicolas, J.,
Peubey, C., Radu, R., Schepers, D., Simmons, A., Soci, C., Abdalla, S., Abellan, X., Balsamo, G.,
Bechtold, P., Biavati, G., Bidlot, J., Bonavita, M., De Chiara, G., Dahlgren, P., Dee, D., Diamantakis, M.,
Dragani, R., Flemming, J., Forbes, R., Fuentes, M., Geer, A., Haimberger, L., Healy, S., Hogan, R. J.,
Hólm, E., Janisková, M., Keeley, S., Laloyaux, P., Lopez, P., Lupu, C., Radnoti, G., de Rosnay, P., Rozum,
I., Vamborg, F., Villaume, S., and Thépaut, J.-N.: The ERA5 global reanalysis, 146, 1999-2049,
https://doi.org/10.1002/qj.3803, 2020.

Huang, J. and Jaeglé, L.: Wintertime enhancements of sea salt aerosol in polar regions consistent
with a sea ice source from blowing snow, Atmos. Chem. Phys., 17, 3699-3712, 10.5194/acp-17-3699-
2017, 2017.

Jaeglé, L., Wood, R., and Wargan, K.: Multiyear Composite View of Ozone Enhancements and
Stratosphere-to-Troposphere Transport in Dry Intrusions of Northern Hemisphere Extratropical
Cyclones, 122, 13,436-413,457, https://doi.org/10.1002/2017JD027656, 2017.

Jaeglé, L., Quinn, P. K., Bates, T. S., Alexander, B., and Lin, J. T.: Global distribution of sea salt
aerosols: new constraints from in situ and remote sensing observations, Atmospheric Chemistry and
Physics, 11, 3137-3157, 10.5194/acp-11-3137-2011, 2011.

Jiang, B., Xie, Z., Lam, P. K. S., He, P., Yue, F., Wang, L., Huang, Y., Kang, H., Yu, X., and Wu, X.:
Spatial and Temporal Distribution of Sea Salt Aerosol Mass Concentrations in the Marine Boundary
Layer From the Arctic to the Antarctic, 126, e2020JD033892, https://doi.org/10.1029/2020JD033892,
2021.

Kawaguchi, Y., Hoppmann, M., Shirasawa, K., Rabe, B., and Kuznetsov, I.: Dependency of the drag coefficient on boundary layer stability beneath drifting sea ice in the central Arctic Ocean, Scientific reports, 14, 15446, 10.1038/s41598-024-66124-8, 2024.

Lai, Z. L., Cheng, Z. Z., Lata, N. N., Mathai, S., Marcus, M. A., Mazzola, M., Mazzoleni, C., Gilardoni, S., and China, S.: Chemical Composition and Mixing State of Wintertime Aerosol from the European Arctic Site of Ny-Ålesund, Svalbard, Acs Earth and Space Chemistry, 10.1021/acsearthspacechem.5c00175, 2025.

Lan, H. T., Guo, D. L., Hua, W., Pepin, N., and Sun, J. Q.: Evaluation of reanalysis air temperature and precipitation in high-latitude Asia using ground-based observations, International Journal of Climatology, 43, 1621-1638, 10.1002/joc.7937, 2023.

Li, L., Li, M., Huang, Z., Gao, W., Nian, H., Fu, Z., Gao, J., Chai, F., and Zhou, Z.: Ambient particle characterization by single particle aerosol mass spectrometry in an urban area of Beijing, Atmospheric Environment, 94, 323-331, https://doi.org/10.1016/j.atmosenv.2014.03.048, 2014.

McErlich, C., McDonald, A., Renwick, J., and Schuddeboom, A.: An Assessment of Southern Hemisphere Extratropical Cyclones in ERA5 Using WindSat, 128, e2023JD038554, https://doi.org/10.1029/2023JD038554, 2023.

McPhee, M. G.: TURBULENT HEAT AND MOMENTUM-TRANSFER IN THE OCEANIC BOUNDARY-LAYER UNDER MELTING PACK ICE, Journal of Geophysical Research-Oceans, 88, 2827-2835, 10.1029/JC088iC05p02827, 1983.

Pohl, C., Rey, M., Jensen, D., and Kerth, J.: Determination of sodium and ammonium ions in disproportionate concentration ratios by ion chromatography, Journal of Chromatography A, 850, 239-245, https://doi.org/10.1016/S0021-9673(99)00002-3, 1999.

Priestley, M. D. K. and Catto, J. L.: Improved Representation of Extratropical Cyclone Structure in HighResMIP Models, 49, e2021GL096708, https://doi.org/10.1029/2021GL096708, 2022.

Ranjithkumar, A., Duncan, E., Yang, X., Partridge, D. G., Lachlan-Cope, T., Gong, X. D., Nishimura, K., and Frey, M. M.: Direct observation of Arctic Sea salt aerosol production from blowing snow and modeling over a changing sea ice environment, Elementa-Science of the Anthropocene, 13, 10.1525/elementa.2024.00006, 2025.

Riley, J. P. and Chester, R. J. A. P.: Introduction to marine chemistry, 1971.

Tulger Kara, G. and Elbir, T.: Seasonal and spatial variability in the accuracy of hourly ERA5 and MERRA-2 reanalysis datasets: A 14-year comparison with observed meteorological data in Türkiye, Atmospheric Research, 325, 108233, https://doi.org/10.1016/j.atmosres.2025.108233, 2025.

Wainwright, C. D., Pierce, J. R., Liggio, J., Strawbridge, K. B., Macdonald, A. M., and Leaitch, R. W.: The effect of model spatial resolution on Secondary Organic Aerosol predictions: a case study at Whistler, BC, Canada, Atmos. Chem. Phys., 12, 10911-10923, 10.5194/acp-12-10911-2012, 2012.

Xu, Q., Zhang, S., Zhang, W., Jin, L., Tanaka, K., Haraguchi, H., and Itoh, A.: Amperometric detection studies of Nafion/indium hexacyanoferrate film for the determination of electroinactive cations in ion chromatography, Fresenius' journal of analytical chemistry, 367, 241-245, 10.1007/s002160000341, 2000.

Yan, J., Jung, J., Lin, Q., Zhang, M., Xu, S., and Zhao, S.: Effect of sea ice retreat on marine aerosol emissions in the Southern Ocean, Antarctica, Science of The Total Environment, 745, 140773, https://doi.org/10.1016/j.scitotenv.2020.140773, 2020.

Yan, J., Jung, J., Zhang, M., Xu, S., Lin, Q., Zhao, S., and Chen, L.: Significant Underestimation of Gaseous Methanesulfonic Acid (MSA) over Southern Ocean, Environmental science & technology, 53,

13064-13070, 10.1021/acs.est.9b05362, 2019.